# Omega-3 and Omega-6 Fatty Acids in Poultry Nutrition: Effect on Production Performance and Health

**DOI:** 10.3390/ani9080573

**Published:** 2019-08-18

**Authors:** Mahmoud Alagawany, Shaaban S. Elnesr, Mayada R. Farag, Mohamed E. Abd El-Hack, Asmaa F. Khafaga, Ayman E. Taha, Ruchi Tiwari, Mohd. Iqbal Yatoo, Prakash Bhatt, Sandip Kumar Khurana, Kuldeep Dhama

**Affiliations:** 1Department of Poultry, Faculty of Agriculture, Zagazig University, Zagazig 44511, Egypt; 2Department of Poultry Production, Faculty of Agriculture, Fayoum University, Fayoum 63514, Egypt; 3Forensic Medicine and Toxicology Department, Faculty of Veterinary Medicine, Zagazig University, Zagazig 44511, Egypt; 4Department of Pathology, Faculty of Veterinary Medicine, Alexandria University, Edfina 22758, Egypt; 5Department of Animal Husbandry and Animal Wealth Development, Faculty of Veterinary Medicine, Alexandria University, Behira, Rasheed, Edfina 22758, Egypt; 6Department of Veterinary Microbiology and Immunology, College of Veterinary Sciences, UP Pandit Deen Dayal Upadhayay Pashu Chikitsa Vigyan Vishwavidyalay Evum Go-Anusandhan Sansthan (DUVASU), Mathura 281001, Uttar Pradesh, India; 7Division of Veterinary Clinical Complex, Faculty of Veterinary Sciences and Animal Husbandry, Jammu and Kashmir, Srinagar 190006, India; 8Teaching Veterinary Clinical Complex, College of Veterinary and Animal Sciences, Govind Ballabh Pant University of Agriculture and Technology, Pantnagar 263145, (Udham Singh Nagar), Uttarakhand, India; 9ICAR-Central Institute for Research on Buffaloes, Sirsa Road, Hisar 125 001, Haryana, India; 10Division of Pathology, ICAR-Indian Veterinary Research Institute, Izatnagar, Bareilly 243122, Uttar Pradesh, India

**Keywords:** omega-3, omega-6, fatty acid, nutrition, performance, antioxidant, egg and meat quality, fertility, immunity, health

## Abstract

**Simple Summary:**

In this review, we discuss previous studies, state-of-the-art technology, and the potential implications of utilizing omega-3 and omega-6 fatty acids in poultry diets, as well as the application of these fatty acids in the poultry industry for improving poultry production and health. Essential roles are played by these fatty acids in development and metabolism, growth and productive performance, immune response and anti-oxidative properties, improving meat quality, bone growth and development, and improving fertility rates and semen quality.

**Abstract:**

Omega-3 (ω-3) and omega-6 (ω-6) fatty acids are important components of cell membranes. They are essential for health and normal physiological functioning of humans. Not all fatty acids can be produced endogenously owing to the absence of certain desaturases; however, they are required in a ratio that is not naturally achieved by the standard diet of industrialized nations. Poultry products have become the primary source of long-chain polyunsaturated fatty acids (LC-PUFA), with one of the most effective solutions being to increase the accretion of PUFAs in chicken products via the adjustment of fatty acids in poultry diets. Several studies have reported the favorable effects of ω-3 PUFA on bone strength, bone mineral content and density, and semen quality. However, other studies concluded negative effects of LC-PUFA on meat quality and palatability, and acceptability by consumers. The present review discussed the practical application of ω-3 and ω-6 fatty acids in poultry diets, and studied the critical effects of these fatty acids on productive performance, blood biochemistry, immunity, carcass traits, bone traits, egg and meat quality, and semen quality in poultry. Future studies are required to determine how poultry products can be produced with higher contents of PUFAs and favorable fatty acid composition, at low cost and without negative effects on palatability and quality.

## 1. Introduction

Fatty acids, especially essential fatty acids, are gaining importance in poultry feeding systems not only for improving the health and productivity of birds, but also because of our health-conscious society that prefers properly balanced diets to minimize adverse health issues [1,2,3]. Among various fatty acids, omega-6 (ω-6) and omega-3 (ω-3) fatty acids are proving indispensable in a properly maintained ratio for numerous biological [4,5], physiological [2], developmental [6], reproductive [7], and beneficial health functions [3,8,9]. Adequate supplementation of poultry diets with novel and beneficial feed additives or supplements is gaining importance as it significantly improves overall poultry production and performance as well as safeguards the health of birds [10,11,12,13]. In poultry production, the advantages of using oils in diets involves a reduction of feed dust and improvement in hydrolysis and absorption of the lipoproteins that supply fatty acids [14]. In addition, oils are the main source of energy for the birds and have the highest caloric value among all dietary nutrients. They can also enhance the absorption of fat-soluble vitamins, increase diet palatability, and improve the utilization of the consumed energy. Moreover, the rate of food passage through the gastrointestinal tract can be reduced, with subsequent better absorption of all dietary nutrients [15].

In the diets of humans, ω-6 and ω-3 are essential fatty acids. However, considerable modification in dietary patterns has resulted in alterations of the consumption of such fatty acids, with subsequent elevation in the consumption of ω-6fatty acids and a marked decrease in the consumption of ω-3 fatty acids. This modification has led to an imbalance in the ω-6/ω-3 ratio, which at 20:1 now differs considerably from the original ratio (1:1). Therefore, dietary supplements of foodstuffs such as eggs and meat are a clear alternative to increase the daily consumption of ω-3 to meet the recommended doses [16]. Foods that provide ω-6 fatty acids include soybean, palm, sunflower, and rapeseed oils, whereas foods that provide ω-3 fatty acids include certain nuts, and plant and fish oils [17,18,19,20]. Omega-9 fatty acids are not essential and are found in olive oil and animal fat [21]. Fish oil (FO) includes two types of ω-3 fatty acids: docosahexaenoic acid (DHA) and eicosapentaenoic acid (EPA). Certain vegetables, nuts, and seed oils include alpha-linolenic acid (ALA), which can be converted to DHA and EPA in the body, with linseed oil containing more than 50% ALA [22].

High doses of omega sources in the diet may have deleterious effects on humans, such as increased bleeding risk and higher levels of low density lipoprotein (LDL) cholesterol. Omega-3 fatty acids may influence heart rates. The consumption of high rates of fatty acids (e.g., ω-6) has been linked with a higher occurrence of health problems, such as type 2 diabetes, obesity, and coronary artery diseases [23]. Maintaining a proper ratio of ω-3 and ω-6 fatty acids not only improves performance, but also prevents these health risks. Devising the correct ratios requires the addition of oils having appropriate ω-3 and ω-6 fatty acid levels [24]. Omega-3 fatty acids EPA and DHA have shown many health benefits; they are helpful in fetal development and cardiovascular function, and prevent Alzheimer’s disease [25]. In addition, they play a role in modulating immunity [26,27]. The ratio of n-6:ω-3 fatty acids also plays an important role in the immune response, production performance of broilers and designing meat enriched with ω-3 polyunsaturated fatty acids (PUFAs) [26,27]. The addition of FO to the poultry diet may yield poultry products (such as eggs and meat) that are enriched with ω-3, such as EPA and DHA. Additionally, FO is more effective than other vegetable oils [28]. However, many aspects of essential fatty acids are still unknown and their diverse functions and importance in health and production should be explored. The objective of the present review is to assess the influence of dietary ω-6 and ω-3 PUFAs on the productive performance, antioxidant properties, immunity, carcass traits, bones, egg and meat quality, and semen quality of poultry, as well as their limitations in the poultry industry.

## 2. Beneficial Applications of ω-3 and ω-6 Oils in Poultry

Adding ω-3 and ω-6 fatty acids to the diet has become more important recently [3,29]. For at least the past three decades, studies on the beneficial activities of long-chain PUFAs (LC-PUFAs) in biological processes have been conducted. Dietary intervention with ω-3 may influence chicken immunity and lead to the production of poultry products with health benefits for the consumer [30]. Therefore, research on broiler chickens has focused on the functional action of different LC-PUFA forms and their dietary levels on the metabolism of lipids in birds. Other than the LC-PUFA source, high levels of fatty acids, particularly of the ω-3 family, lead to accelerated lipid oxidation when broiler chickens are under oxidative stress because of genetic selection [31].

Using PUFAs in poultry diets significantly reduces the cholesterol and total lipid content in the blood and egg yolk. Several studies have been conducted to minimize the harmful effects of triglycerides and total cholesterol in poultry products (edible parts). Ahmad et al. [32] reported that the cholesterol content of eggs was decreased when birds were fed a diet supplemented with ω-3 fatty acids. Moreover, increasing dietary levels of FO and milled flaxseed improved the concentration of linoleic acid (LA), EPA, and DHA in the yolk, and the fatty acid deposition from FO was found to be two times greater than that from milled flaxseed when fed at the same dietary levels [33].

Designer eggs offer balanced ratios of PUFA: SFA (1:1) or ω-6/ω-3 PUFA (1:1). Omega-3 UFAs are important nutritional factors that modulate immune functions and are of great importance for nervous system development and for lowering blood platelet aggregation and the incidence of thrombosis, hypertension, and atherosclerosis, and have anti-tumor, anti-inflammatory, and cardioprotective effects [34]. The content of ω-3 fatty acids in eggs can be increased by supplementing the diets of laying hens with certain dietary supplements, such as flaxseed, fish oil, safflower oil, linseed, fish meal, or algae. Omega-3 fatty acids can be introduced to the human consumer’s body through these designer eggs; they play an important role in the maintenance of the normal functioning of the body in that they protect the body from cardiovascular problems such as heart attacks. In addition, they can replace fish products in consumer diets [35]. Ebeid et al. [36] stated that hens fed a diet containing different concentrations of ω-3 PUFA showed a linear decrease and increase (*p* < 0.05) in egg yolk content of ω-6 PUFA and ω-3 PUFA, respectively, compared to the control hen group. The levels of ALA, EPA, DHA, and docosapentaenoic acid (DPA) were higher in the egg yolks of laying hens fed linseed meal and fish oil as dietary supplements than in the un-supplemented hen group [37]. Similarly, ALA was higher in the egg yolk of hens fed diets that contained ω-3 fatty acid dietary supplements than in the control bird group [32]. It has been previously concluded that laying hens can synthesize EPA and DHA from ALA during metabolic processes if ALA is present in adequate quantities [38,39]. However, in mammals, the synthesis rates of DHA from ALA are low compared to the dietary intake and tissue demand, with the estimation of percent conversion of ALA to DHA differing widely, ranging from 0% to 9.2%, supporting the conclusion that DHA synthesis from ingested ALA is not an efficient process in humans [40]. Moreover, metabolization of ALA in vivo is not adequate to improve meat quality in ω-3 LC-PUFA, and direct supplementation of the diet with ω-3 LC-PUFA is a better alternative to modulate an increase in beneficial fatty acids of broiler meat [41]. The efficiency of ALA conversion to ω-3 LC-PUFA derivatives and deposition in peripheral tissues might not be sufficiently high to improve the nutritional value of muscle. Because there is competition among the enzymes involved in the elongation and desaturation of both LA and ALA, high amounts of LA suppress the conversion of ALA to EPA or DHA; therefore, an optimal intake of LA relative to ALA is crucial for normal metabolism [42].

Regarding egg production performances, Buitendach et al. [43] investigated the effects of dietary fatty acid saturation on the production performance of laying hens at end-of-lay (58–74 weeks of age). These authors reported no significant differences in hen-day egg production, egg weight, egg output, feed efficiency, and body weights at end-of-lay. Similar results were reported by Cachaldora et al. [44,45] who concluded that dietary fatty acid saturation had no significant effects on the egg production performance of layer hens. In contrast, Shang et al. [46] stated that body weight gain, rate of egg production, egg weight, and feed efficiency decreased linearly with an increase in dietary fatty acid unsaturation levels during the 8-week experimental period between 40 and 48 weeks of age. Yin et al. [47] reported a decrease in egg and body weights with an increase in dietary UFAs at 50–58 weeks of age. This decrease in the performance parameters of hens as recorded by Shang et al. [46] and Yin et al. [47] is mostly attributed to the fact that these authors used conjugated linoleic acid (CLA) at higher inclusion levels (up to 7.8%) to enhance the UFA profile of their experimental diets. CLA causes weight loss in humans [47]; therefore, it appears that this specific type of unsaturated fatty acid has a similar negative effect on the body weights of laying hens and consequently on egg production and egg size.

## 3. Improved Growth and Productive Performance

Growth and production performance of poultry are improved by supplementation of fatty acids or their sources. The supplementation of fats and oils (as an omega source) in limited amounts leads to better utilization of feed and energy, with subsequent improvement in growth and performance [48]. The body mass and percentage body mass gain of quails was improved via dietary supplementation of sunflower and soyabean oil for a 12-week period [49]. The feed conversion ratio (FCR) and growth performance of broilers were improved via dietary supplementation with sunflower and canola oil [14]. Smith et al. [50] stated that the supplementation of animal fat, corn oil, fish oil, and a blend of vegetable and animal oils did not affect the feed intake, but positively influenced the body gain and FCR in heat-stressed broilers compared to heat-stressed un-supplemented broilers. Jalali et al. [51] found that the addition of soybean oil (high in ω-3 PUFA) significantly (*p* < 0.05) improved the FCR and body weight gain of broilers during the total and growth rearing periods. Abdulla et al. [52] found that the supplementation of soybean oil in broiler diets increased the body weight and weight gain at 6 weeks of age compared with the supplementation of LO (*p* < 0.05).

Fébel et al. [53] noticed that the difference in diets supplemented with sunflower oil (SO) or lard was not significant for the growth performance of broilers. Supplementation of fish oil in poultry diets had no influence on feed consumption, live weight, or weight gain [48] compared with the control diet (without fat). Ebeid et al. [54] reported that dietary ω-3 PUFA in Japanese quail had no adverse effects on the growth performance, such as the final body weight, feed intake, or FCR. Raj Manohar and Edwin [55] declared that dietary ω-3 PUFA in quails had a significant (*p* < 0.01) influence on body weight gain and non-significant differences in feed intake and feed efficiency. In addition, Qi et al. [56] demonstrated that decreasing dietary ω-6/ω-3 improved FCR, with the best result obtained from the diet with 5:1 ω-6/ω-3. Puthpongsiriporn and Scheideler [57] reported that four dietary ratios (17:1, 8:1, 4:1, and 2:1) of LA to ALA had no significant effect on the body weight of chickens over 16 weeks. However, Ayerza et al. [58] observed a marked reduction in the body weight and FCR of broilers fed on chia and flaxseed (rich in ALA), which may be attributed to one or more of the anti-nutritional factors present in flaxseed. Crespo and Esteve Garcia [59], Newman et al. [60], and Ferrini et al. [61] reported that the digestibility of fat increased with increasing unsaturation; therefore, the effect of the type of fat on feed efficiency could reflect the degree of unsaturation. This is consistent with the improvement of growth performance noted by Zollitsch et al. [62], Huo et al. [63], and Lopez-Ferrer et al. [48] with increasing content of UFA.

Generally, a dietary supplementation of a PUFA-enriching ingredient has an improvement effect on live weight, weight gain, and FCR of poultry. However, no adverse effects on feed intake have been reported in almost all designed studies to date.

## 4. Improved Immune Response and Anti-Oxidative Properties

Fatty acid supplementation affects both immune and oxidative status in poultry. It can modulate immune system through both cellular and humoral immune responses. Proliferation, maturation, function and cytokine production of lymphocytes, heterophils and splenocytes are influenced by omega fatty acids, besides antibody production like IgM and IgG (Table 1). Similarly, neutralization of oxidants and an increase of antioxidants level directly or indirectly by fatty acids minimizes the risk of oxidative stress. The immune response and oxidative mechanisms are interlinked and affect one another, hence modulation of one can impact the other. The supplementation of natural antioxidants has become a pressing research topic [64,65,66]. Different studies have confirmed the many favorable effects of dietary ω-3 PUFA, including anti-oxidative properties and lipid peroxidation, as well as immune response effects [36,67]. For instance, dietary ω-3 PUFA can modulate the immune response in poultry [68]. Ebeid et al. [36] found that dietary FO supplementation below a level of 35 g/kg in the diet induced antibody titers in hens. The levels of antibodies were higher in laying hens fed oils (FO or LO) rich in ω-3 PUFAs than those in laying hens fed oil rich in ω-6 PUFAs (maize oil) [69]. Ebeid et al. [54] stated that dietary ω-3 PUFA (FO or LO) had a positive influence on humoral immunity (*p* ≤ 0.05) at 42 days of age as measured by antibody titers against Newcastle disease virus (NDV) compared to the control diet. Al-Khalifa et al. [70] declared that SO replaced by fish oil at low levels showed no evidence of adverse effects on the immune function of broilers. Jameel et al. [71] showed that chicks fed a diet supplemented with FO had significantly higher (*p* ≤ 0.05) antibody titers and percentages of spleen and bursa than that of the control group.

One of the earliest studies on the effects of fatty acid supplementation on immune tissues was by Fritsche et al. [72]. They noted that the supplementation of diets rich in ω-3 fatty acids to chicks decreased levels of arachidonic acid (AA) (C20:4n-6) in the serum and immune tissues by 50–75%. However, the levels of EPA (C20:5n-3) and DHA (C20:6n-3) increased [72], suggesting an influence on the immune system. During the early stages of life, ω-3 and ω-6 fatty acids are more important for immunity in chicks as they play a role in cellular and humoral immunity, and are regulators of inflammation [1,73]. They determine the immunoglobulin G (IgG) content of chicks produced by maternal hens, which are essential for passive immunity [74]. An inflammatory role for these fatty acids in delayed-type hypersensitivity has also been reported [75]. Further, they help in maintaining membrane integrity, thus preventing pathogen entry or infection.

Wang et al. [68] noted that the LA to ALA ratio may influence IgG-receptor activity in yolk sac membranes and thereby influence the maternal-embryo transfer of yolk IgG. Adding fish oil to broiler diets significantly induced antibody titers for the LaSota vaccine at 35 days of age after vaccination against Newcastle disease because of ω-3, which plays a role in the production of immunomodulators (leukotriene and prostaglandin). In addition, fish oil has the capacity to modulate the production of cytokines via signal transduction and lymphocytes in a population of immune cells [76]. Al-Mayah [76] showed that chickens fed a diet supplemented with fish oil at a level of 50 g/kg showed a higher production of antibodies (IgM and IgG) and globulins in the serum and maintained immune function after vaccination compared to the control group.

A moderate intake of ω-3 PUFAs enhances anti-oxidative properties, such as glutathione peroxidase (GSH-Px) activity, in laying hens [36] and decreases lipid peroxidation in abdominal fat and serum [36,67]. Ebeid et al. [54] reported that adding FO and LO at a level of 20 g/kg to Japanese quail diets significantly increased both the total antioxidant capacity and GSH-Px activity, and decreased the thiobarbituric acid reactive substances in the serum compared with the negative control. SO-enriched diets led to a reduction in the deposition of abdominal fat [77]. Additionally, adding SO and LO to the diet of birds led to a greater decrease in abdominal fat deposition than that observed after adding olive oil and tallow [67]. The addition of SO in broiler diets significantly increased (*p* < 0.05) the relative weight of the abdominal fat pad [51], whereas the abdominal fat of broilers was decreased with fish oil [64,78]. Diets high in ω-3 fatty acids increased the incorporation of these fatty acids into tissue lipids, leading to oxidative stress in cells [30]. A diet enriched in ω-3 PUFA improved the gene expression of lipin-1, which regulates triglyceride synthesis, in chicken abdominal fat [79]. Ibrahim et al. [26] stated that FO and LO supplementation had a low but significant effect (*p* ≤ 0.05) on the malondialdehyde (MDA) concentration in broilers. Reducing the ratios of ω-6:ω-3 PUFAs were found to be linked to a significant (*p* ≤ 0.05) induction of glutathione S transferase (GSH-ST). Moreover, superoxide dismutase, GSH-ST, and cardiac GSH-Px activities were augmented in the ω-3 PUFA-rich treatment, and MDA was reduced [79]. Omega-3 PUFAs have shown beneficial immune responses in infectious bursal disease challenged broilers [80]. However, despite the noted improvements, these fatty acids must be evaluated and properly monitored for ratios to prevent adverse effects on immune status [70].

## 5. Improving Egg Quality and Nutritional Value of Eggs

The nutraceutical value and health benefits of eggs can be enhanced by adapting appropriate feeding strategies in poultry as well as by developing designer eggs [10,12,20]. These improve the quality and quantity of eggs [33]. Eggs are not naturally rich in ω-3 PUFA; therefore, ω-3 PUFA supplementation in poultry rations is required to obtain enriched ω-3 PUFA eggs [81,82]. Designer eggs are enriched in ω-3 fatty acids for beneficial health effects in human nutrition [12,83]. Designer eggs offer balanced ratios of PUFA: SFA (1:1) or ω-6/ω-3 PUFA (1:1) and provide more than 600 mg of ω-3 PUFA [34]. The content of ω-3 fatty acids in eggs can be increased by supplementing the diets of laying hens with certain dietary supplements, such as groundnut oil, fish oil, safflower oil, linseed, fish meal, or algae [15,16,17,18,58]. Omega-3 fatty acids include EPA, DPA, DHA, and linolenic acid (LNA), whereas AA and LA are examples of ω-6 fatty acids. Omega-3 fatty acids can be introduced to the body through designer eggs [35]. Omega 3-PUFAs serve as good fats for human health, therefore increasing PUFA contents in the egg yolk helps to decrease the bad cholesterol content [84]. The stability of ω-3 PUFAs can be improved by vitamin E and/or organic selenium, which reduces oxidation in raw eggs; thus, these confer protective effects during the marketing, storage, and cooking of ω-3 enriched eggs [85,86].

Meluzzi et al. [87] reported that the key ω-3 and ω-6 PUFAs are LNA and LA, respectively. LNA is metabolized inside the body to EPA, DHA, and DPA, whereas LA is metabolized to AA. LNA was higher in the egg yolk of hens after feeding them diets that contained ω-3 fatty acid dietary supplements than those of the control birds [88].

Ceylan et al. [89] evaluated the effect of dietary supplementation on two levels (15 g/kg and30 g/kg diet) of SO, rapeseed oil, and LO for 12 weeks in laying hens. They concluded that egg production, egg weight, feed intake, FCR, and live weight were not significantly affected by the treatments. However, hens receiving SO produced less intensively colored egg yolks than those receiving other oils in their diet (*p* < 0.01). Moreover, the composition of fatty acid in egg yolks was significantly (*p* < 0.01) affected by the treatment, whereas the cholesterol content was not influenced. There was a significant (*p* < 0.05) interaction between fat source and the level of inclusion in the diet, and LNA content was increased when hens were fed diets with linseed and rapeseed oil (30 g/kg diet). In contrast, da Silva Filardi et al. [90] studied the effects of the dietary inclusion (for 12 weeks) of different fat sources (cottonseed oil, soybean oil, lard, SO, or canola oil) on egg quality, and egg yolk lipid profiles. The different fat sources did not affect eggshell quality; however, the lipid profile of the egg yolk changed based on dietary fat sources. Optimal changes were considered to be lower levels of SFA and LA, and higher levels of ALA and DHA. Such changes were promoted by the addition of different fat sources, particularly canola oil; however, it did not enhance the egg content of PUFAs.

## 6. Improving Meat Quality

In human diets, there is a marked reduction in ω-3PUFA and an imbalance in the ratio of ω-6/ω-3 PUFA. Currently, the ratio of ω-6 to ω-3 fatty acids is approximately 10 to 20:1 rather than the recommended ratio (1 to 4:1). The decrease in ω-3PUFA consumption is due to the low intake of sea fish, which are the major source of ω-3 PUFA. An accepted solution for this situation could be based on the production of suitable functional foods with adjusted PUFA content, which is generally accepted to confer nutritional effects and beneficial physiological properties. The enrichment of poultry meat with ω-3 PUFA may provide an excellent alternative source for such acids in the human diet because of their relative availability [87]. Schiavone et al. [91] illustrated that the content of lipids, protein, and moisture in breast meat was not significantly affected by the addition of fish oil to the diet of the Muscovy duck. Additionally, Ebeid et al. [54] reported that adding n-3 PUFA to Japanese quail diets had no significant influence on the content of crude protein, ash, and dry matter in the meat, whereas the addition of ether extract significantly influenced these parameters. Additionally, the physical traits of the meat except for the water-holding capacity were not significantly influenced when ω-3PUFA was added to the diet. Though the fatty acid composition of meat is influenced by ω-3 PUFA supplementation or their sources in diet, meat quality parameters like meat pH, tenderness, grilling loss, toughness, and juiciness are not affected [48,54,87,91]. This can be exploited for designing functional foods with adjusted PUFA and having no differences in palatability.

In broiler diets, replacing soybean oil with LO along with the addition of pomegranate peel extract enriched muscle meat with antioxidants and ω-3 and improved broiler immunity and their serum lipid profile [92]. Also, natural antioxidants, especially those extracted from herbal plants, have greater potential for increasing the stability, palatability, and shelf-life of meat products [93,94]. The meat quality of broilers improved with fish oil supplementation in the diet [78]. Inclusion of fish oil (FO) and different fat sources [linseed oil (LO), rapeseed oil (RO), sunflower oil (SO)] for providing different PUFA (ω-3 and ω-6 PUFA) in diets and their deposition into the eggs’ fat revealed that smaller proportions of FO resulted in lower values of saturated and higher values of ω-6 FA contents. Replacing FO with LO showed the lowest turn down of its derivatives by elongation and desaturation and an increase in the total ω-3 FA in the form of linolenic acid [95]. The use of LO as ground or whole flaxseed before slaughter is recommended to broiler breeders and producers as a feeding strategy to optimize ω-3 enrichment, without compromising poultry performance [96]. The fat and cholesterol content in poultry meat may decrease because of dietary supplements, such as ω-3 PUFA, and high PUFA concentrations in the diet (addition of vegetable oils) decrease the storage stability of meat [97].

A proper ratio of ω-6:ω-3 fatty acids is essential for maintaining health, oxidative balance and quality of meat. Recently, Konieczka et al. [8] found that feeding birds with a diet containing a PUFA ω-6:ω-3 ratio exceeding the recommended levels resulted in damage to the intestinal epithelial cells. Further, low PUFA ω-6:ω-3 ratio diets increased MDA in tissues including the meat. This can affect meat quality owing to peroxidative changes [8]. Hence, these authors recommended balanced supplementation to prevent oxidative damage and loss of meat quality. Similarly, Kalakuntla et al. [6] noted that the supplementation of ω-3 PUFA-rich oil sources in the broiler diet during starter and finisher phases can affect fatty acid composition, quality, and organoleptic characters of broiler chicken meat. At 2% and 3% addition levels, mustard oil, fish oil, and LO improved ω-3 PUFA levels and sensory attributes such as the appearance, flavor, juiciness, tenderness, and overall acceptability of meat; however, due to an increase of thiobarbituric acid-reacting substances, the meat quality might be compromised as it might cause oxidative damage to meat.

The content of ω-3 fatty acids in poultry meat, especially as EPA and DHA, can be readily improved by increasing the levels of ω-3 PUFA in poultry diets via the inclusion of oily fish by-products [98]. Qi et al. [56] concluded that substituting ω-3 for ω-6 in the diets of chickens resulted in a significant effect on the subcutaneous and intramuscular fat content and on meat quality (color and tenderness).

Unfortunately, although poultry meat is considered one of the main potential sources of ω-3 LC-PUFA for humans, particularly in developed countries [99,100], there are some disadvantages related to meat oxidative stability. LC-PUFAs are very susceptible to oxidation, producing off-flavors and odors in meat that are often associated with a fishy flavor [101]. This oxidative instability can influence meat quality and, consequently, reduce acceptability to consumers [41,102,103]. Oken et al. [104] concluded that the supplementation of chicken diets with fish-derived products led to unacceptable odors in the product, which has restricted the adoption of this strategy [105]. Vegetable sources such as LO may clearly increase the ω-3 PUFA content in the form of ALA, which is the precursor of the entire ω-3 family [42].

Conclusively, dietary supplementation of ω-3 fatty acids in poultry diet, particularly in the form of EPA and DHA, can improve various parameters of meat quality. However, LC-PUFAs are extremely susceptible to oxidative deterioration, resulting in off-flavors and odors, which adversely affect acceptability to consumers, especially when fish-derived products are used.

## 7. Effects of Dietary ω-3 and ω-6 Fatty Acids on Bones

The ω-3 and ω-6 fatty acids or their sources like fish oil, linseed oil, soybean oil and palm oil have bearing effects on mineral metabolism and hence promote bone formation, growth and development. Fat supplementation in the diet influences mineral metabolism, especially calcium, zinc, and magnesium [106], because of insoluble soap formation between these minerals and fatty acids during digestion, which makes them unavailable [107]. This can affect mineral retention, and influence bone and eggshell quality in birds. Dietary lipids play a remarkable role in the growth, development, and formation of bones [106,107,108]. Sun et al. [108] reported that dietary fish oil supplementation led to significantly higher bone mineral density in the proximal tibia and distal femur than supplementation with maize oil. Some studies have reported non-significant correlations between fatty acid supplementation and bone characteristics. Baird et al. [109] reported that feeding laying chickens a diet high in ω-3 PUFA did not have a significant effect on bone morphological characteristics, bone mineral content, or bone mineral density. However, there are many studies that can prove interrelation of fatty acid supplementation and bone growth and development. Ebeid et al. [36] declared that using ω-3 PUFA in Japanese quail diets improved the tibia bone and morphological characteristics and, in quails fed diets supplemented with FO and LO at a level of 20 g/kg diet, there was increased tibia bone wall thickness, tibia diameter, and the percentage of tibia ash and tibia bone breaking strength compared with that in quails fed the control diet. Abdulla et al. [52] clarified that chicks fed a diet supplemented with LO had non-significantly higher ash percentage, tibia weight, and bone-breaking strength than those fed diets supplemented with SO and palm oil. In addition, ω-3 PUFA may improve bone health by inducing calcium absorption in the gut and inducing osteoblast activity and differentiation, decreasing osteoclast activity, and stimulating the deposition of minerals in developing bones [110]. Reproducible and consistent beneficial effects of ω-3 fatty acids have been observed for bone/joint diseases and bone metabolism [111]. Recently, the importance of yolk as a mineral source for chicks and possible alterations via interventions strategies for future usage has been analyzed [112]. The amount of minerals in yolk reflects that the uptake content and enrichment can have beneficial effects [113]. In-ovo supplementation of minerals has improved bone properties and development in hatchlings and mature broilers [113,114,115].

## 8. Improved Fertility Rates and Semen Quality

Fatty acid supplementation, especially of ω-3 and ω-6, helps in improving fertility, semen quality and quantity. Kelso et al. [114] found that dietary fish oil or corn oil supplementation to chickens at a level of 50 g/kg in their diet led to significantly higher (96%) fertility rates than the rates prior to supplementation (89%). Kelso et al. [115] noticed that the supplementation of ALA in male diets led to higher fertility at 39 weeks of age because of the increased ω-3 fatty acid proportion in the phospholipids of sperm. Cerolini et al. [116] reported that dietary FA supplementation can influence spermatozoa traits. Hudson and Wilson [117] stated that the supplementation of menhaden oil at 30 g/kg in the diets of male broiler breeders improved the quality of semen and increased fertility and hatchability. Bongalhardo et al. [118] reported that supplementing cockerel diets with fish oil improved fertility, which was attributed to the lower fatty acid ratio (ω-6:ω-3) in the membrane of spermatozoa that may change the membrane resistance to peroxidative damage or its physical characteristics [119].

Fertility and quality of sperms both have been found to be affected during cryopreservation, and fatty acids act as protectants for sperms. Cryopreservation of semen affects survivability, which to a greater extent is dependent on lipid content in spermatozoa [119]. Blesbois et al. [119] noted a decrease in cholesterol/phospholipid ratio in poultry sperms following cryopreservation and a relation with fluidity, hence affecting survivability. Fatty acids can prevent damage to sperms, whether it be physical (cryopreservation) or chemical (oxidative). Zaniboni and Cerolini [120] stated that the dietary ω-3 LC-PUFA treatment of turkey prevented the negative influence of sperm storage on sperm sensitivity and quality and promoted in-vitro peroxidation and sperm death. Additionally, dietary maize oil supplementation decreased the spermatozoa number per ejaculate by 50% between 26 and 60 weeks of age. Al-Daraji et al. [121] noticed that dietary fish oil supplementation produced the best results for sperm concentration (*p* < 0.05) based on the ejaculate volume, live sperm, total sperm count, and sperm quality factors, followed by flax oil; however, the worst results for these traits were found with treatments of corn oil and SO. Al-Daraji [122] determined that the correlation between the spermatozoa number and glucose concentration in the seminal plasma was highly significant and negative, indicating that the spermatozoa utilized glucose. Additionally, Al-Daraji [123] noticed that spermatozoa used glucose in the seminal plasma for metabolism. Al-Daraji et al. [121] also clarified that dietary SO or corn oil supplementation had a significant effect (*p* < 0.05) on the semen glucose content, alanine aminotransferase activity, and semen protein content, followed by the results for flax oil and fish oil.

A diet supplemented with a moderate ratio of ω-3:ω-6 fatty acids increased DHA and ω-3PUFAs and decreased docosatetraenoic acid and AA in rooster sperm [124]. Sperm motility, progressive motility, membrane functionality, and viability were significantly improved; the testosterone concentration increased; and a higher fertility rate was noted [125]. Feng et al. [7] reported no significant effect on the testis index; however, the spermatogonial development and germ cell layers and gonadotropin-releasing hormone, luteinizing hormone (LH), follicle-stimulating hormone (FSH), and testosterone hormone levels increased. Further, they reported that PUFAs regulate the expression of hormone receptors and steroid acute regulator protein (StAR). PUFAs significantly increased the mRNA levels of all hormone-related genes (GnRHR, FSHR, LHR, and StAR mRNA levels).

An overview of different dietary manipulations for improving nutritional quality of poultry products (egg and meat) is presented in Table 1.

## 9. Conclusions and Future Perspectives

The present review revealed that ω-3 and ω-6 fatty acids could be successfully utilized in poultry feeds to promote immune responses and improve the nutritional value of eggs, meat quality and growth in poultry. Omega-3 fatty acids have anti-inflammatory or inflammation-reducing properties because they can reduce the liberation of cytokines. Omega-6 fatty acids at high levels are associated with an increased prevalence of severe conditions, such as depression and heart disease. However, these fatty acids have a tremendous range of health benefits, including improved cholesterol levels and a reduced occurrence of coronary heart disease. Numerous studies have reported the favorable effects of ω-3 PUFA on bone strength, bone mineral content, and bone mineral density. Furthermore, the content of ω-3 fatty acids in eggs can be increased by supplementing the diets of laying hens with certain dietary supplements, such as groundnut oil, fish oil, safflower oil, linseed, fish meal, or algae. Dietary supplementation with different sources of ω-3 or ω-6 to cockerels improved the semen quality and increased fertility and hatchability. In the present review, we proposed that supplementing poultry diets with different sources of ω-3 and ω-6 fatty acids represented a potential strategy for poultry produced for human consumption. However, some disadvantages were related to meat oxidative stability, where LC-PUFAs were very susceptible to oxidation, resulting in off-flavors and odors in poultry meat, which negatively influence meat quality and acceptability by consumers. Therefore, future studies should investigate how we can produce poultry products with higher contents of PUFAs and favorable fatty acid composition, with low cost and without negative effects on palatability and quality, and subsequently on acceptability by consumers.

## Figures and Tables

**Table 1 animals-09-00573-t001:** Studies showing the different dietary manipulation to improve nutritional quality in poultry products (egg and meat).

Author/s	Supplements	Results
[88]	Linoleic acid to α-linolenic acid ratio from 20:1 to 1:2	Increased ALA from 0.95 to 5.09% of fatty acid, EPA from 0.07 to 0.31% of fatty acids, DHA from 0.44 to 3.54% of fatty acids, and n6/n3 from 9.37 to 1.31.
[124]	Calcium in diet (3.2 and 3.7%), sodium butyrate, probiotic, herbal blend or chitosan	Chitosan increased eggshell thickness, strength, and decreased cholesterol in yolk; herbal extract increased eggshell thickness and had no effect on performance, egg shell quality, fatty acid, and lipid profile.
[125]	* Hemp seed (Cannabis sativa) (100 to 200 g/kg diet)	Increased EPA (0.2 versus 0.9 to 1.2% of fatty acids) and DHA (17.1 versus 39.2 to 47.4% of fatty acids)—decreased n6 to n3 fatty acid ratio (44.9 versus 4.92 to 11.7).
[125]	* Hemp oil (40 and 120 g/kg diet)	Increased EPA (0.3 versus 1.2 to 3.2% of fatty acids), DHA (17.1 versus 40.9 to 48.1% of fatty acids) and ALA (15.8 versus 58.7 to 192% of fatty acids).
[126]	* Hemp seed (Cannabis sativa) (150–250 g/kg diet)	Decreased cholesterol linearly with maximum reduction of 32% (281 versus 191 mg/egg) Increased EPA + DHA from 1.33 mg to 5.76 mg/g of unsaturated fatty acids.
[127]	* Hemp seed (Cannabis sativa) (100 to 300 g/kg diet)	Increased EPA (0 versus 1.12 to 2.66% of fatty acids) and DHA (16.2 versus 41 to 41.3% of fatty acids)—decreased n6 to n3 fatty acid ratio (44.9 versus 4.92 to 11.7).
[127]	* Hemp oil (45 and 90 g/kg diet)	Increased EPA (0 versus 1.35 to 2.13% of fatty acids) and DHA (15.8 versus 39.3 to 43.6% of fatty acids).
[127]	** Medicago sativa (Alfalfa sprouts) 40 g/d	Decreased cholesterol by 9.5% in the egg. Improved isolariciresinol by 220% and daidzein by 173%. Improved EPA by 109%, DHA by 22% and LNA by 22% and also increased many other antioxidant significantly.
[128]	Flax sprouts (Linum usitatissimum) 40 g/d	Decreased cholesterol by 8.7%. Increased isolariciresinol by 142% and daidzein by 327%. Enhanced EPA by 64%, DHA by 91% and LNA by 55%. Increased many other antioxidants significantly.
[129]	*** Fermented buckwheat extract (Fagopyrum esculentum) 16 g/kg diet for 4 weeks	Enriched L-carnitine (13.6%) and GABA (8.4%) in the egg yolk.
[130]	Gynura procumbens (Lour) plant (2.5 to 7.5 g/kg)	Lowered total cholesterol by 12% in the egg yolk.
[131]	Stearidonic acid-enriched soybean oil (50 g/kg diet)	Improved EPA (1 versus 10 mg), DHA (46 versus 84 mg) and total ω-3fatty acids (94 versus 244 mg) per egg yolk.
[131]	**** Fish oil (50 g/kg diet)	Increased EPA (1 versus 56 mg), DHA (46 versus 211 mg) and total n-3 fatty acids (94 versus 340 mg) per egg yolk.
[131]	***** Flaxseed oil (50 g/kg diet)	Improved EPA (1 versus 6 mg), DHA (46 versus 72 mg) and total ω-3fatty acids (94 versus 376 mg) per egg yolk.
[132]	***** Flaxseed oil (10 to 40/kg diet)	Increased EPA (0 versus 0.01 to 0.7% of fatty acid) and DHA (0.74 versus 1.25 to 1.72% of fatty acid) content. Decreased n6/n3 fatty acid (13.3 versus 6.8 to 2.3).
[133]	Feeding of Lacobacillus reuteri (10(6) CFU/mL of bacteria to 1-d-old broiler chickens weekly for 6 weeks)	Enhanced conjugated linoleic acid concentration in eggs (0.16 to 1.1 mg/g fat at 4–5 week of supplementation).
[134]	Pomegranate seed oil, used as a source of punicic acid (5–15 g/kg diet)	Pomegranate seed oil, used as a source of punicic acid (0.5 to 1.5% level. Improved EPA and DHA content in eggs).
[135]	Microalgae (Schizochytrium) powder (5 and 10 g/kg diet)	Increased DHA, but not EPA.
[136]	Addition of PUFA at a ratio of ω-3: ω-6(1:5)	Reduced the cholesterol level of breast meat.
[137]	Increasing doses (0. 3 to 4 g/kg n3-PUFA from microalgae Isochrysis galbana)	Increased n3 long-chain PUFA in egg yolk linearly from 14.7 to 129 mg Transfer efficiency was maximum (53%) at 0.12% level of supplementation with lowest efficiency (28%) at 0.4% level.
[138]	Different n3-PUFA supplementation (0.56% extruded flaxseed, 2.03% Isochrysis galbana, 0.68% fish oil, and 0.44% DHA Gold)	The lowest enrichment efficiency (6%) was observed with flaxseed (α-linolenic acid source). Fish oil, microalgae and DHA Gold had enrichment efficiencies of about 55%, 30% and 45%, respectively.
[139]	Microalgae, Phaeodactylum tricornutum, Nannochloropsis oculata, Isochrysis galbana and Chlorella fusca (25 mg and 250 mg extra n-3 PUFA per 100 g feed)	The highest efficiency of ω-3long-PUFA enrichment was obtained by supplementation of Phaeodactylum and Isochrysis. Yolk color shifted from yellow to a more intense red color with supplementation of Phaeodactylum, Nannochloropsis and Isochrysis.
[140]	PUFA in diet	Meat fat content and composition, meat quality and shelf life, nutritive value.
[141]	Basal diet + 100 mg l-theanine/kg diet; basal diet + 200 mg l-theanine/kg diet; and basal diet + 300 mg l-theanine/kg diet.	Intermediate level of l-theanine (200 mg/kg diet) showed better results in terms of body weight gain (BWG), feed consumption (FC), and feed conversion ratio (FCR). Visceral weight and meat color improved, cholesterol decreased HDL increased, and antioxidant status improved. Higher levels have l-theanine deleterious effects.
[142]	Feeding laying hens with alpha-linolenic acid (ALA) resources [flax (10%), perilla (10%), and Eucommia ulmoides (10%) and eicosapentaenoic acid/docosahexaenoic acid (EPA/DHA) resources (Schizochytrium sp.) (1.5%)	Combination of microalgae and perilla seeds increased ALA from 19.7 to 202.5 mg/egg and EPA + DHA from 27.5 to 159.7 mg/egg. n-3 PUFA enrichment was 379.6 mg/yolk. Combination feeding increased ALA, EPA, and DHA content.
[143]	Feeding linseed (4.5%) + tomato-red pepper mix (1 + 1%)	Linseed decreased palmitic acid (25.41% to 23.43%) and stearic acid (14.75% to 12.52%), no effect on α-Linolenic acid, and increased eicosapentaenoic acid (EPA) (0.011% to 0.047%) and docosahexaenoic acid (DHA) (1.94% to 2.73%). Linseed combined with tomato-red pepper mix did not affect these parameters.
[144]	Feeding wheat-soybean meal basal diet along with sunflower oil (SO), animal oil (AO), linseed oil (LO), or menhaden fish oil (FO)@ 5% (wt/wt)	Significantly lower splenocyte proliferative response to ConA has been noted in chicks fed LO or FO than the chicks fed SO or AO. Significantly lower splenocyte response to PWM has been noted in chicks fed AO, LO, and FO than fed with SO. Significantly lower thymus lymphocyte proliferation in response to ConA in chicks fed AO, LO, and FO than in the chicks fed SO. Proportion of IgM + lymphocytes in spleen increased in both chicks fed LO and FO, however serum IgG concentration increased in FO-fed chicks only. Significant increase in CD8 + T-lymphocytes percentage has been noted in LO-fed chicks.

* One ounce of hulled hemp seeds providing 2.5 g of omega-3. ** Total Omega-3 fatty acids in Medicago sativa equal to 1522 mg (levels per 200-Calorie serving). *** Fermented buckwheat extract providing Omega-3: 0.08 g and Omega-6: 0.96 g (in each100 gm). **** Fish oil contain the omega-3 fatty acids eicosapentaenoic acid (EPA) and docosahexaenoic acid (DHA). ***** One tablespoon of flaxseed oil is providing about 700 milligrams (mg) of EPA and DHA.

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
