# Peer review of "Omega-3 and Omega-6 Fatty Acids in Poultry Nutrition: Effect on Production Performance and Health"

_animals, 2019, doi:10.3390/ani9080573_

Round 1

Reviewer 1 Report

The paper presented for the review contains an overview of the world literature on the nutritional usefulness of n-3 and n-6 fatty acids in poultry. The paper is written well and comprehensibly. The selection of references is correct. Before printing the paper in Animals, I suggest formatting paper according to the requirements of Animals, in particular lines 325-327, Table 1 (143-144), 589-593. 658-660, 696-697, 752-755, 818-828. I suggest correction of computer errors, for example L37 utilizing omega-3 instead of utilizingomega-3, L190 16 weeks instead of 16weeks, L97 removing one dot after [26.27], etc. Please review all the paper in this regard. Introducing a space between the journal name and pages in all References

All remarks contained in the previous review have been included in this paper. When entering the abbreviation for the first time, for example, name FO, enter the full name in brackets or vice versa,                  I suggest L83: Fish oil (FO), Please see all the paper in this respect

The new text entered, marked in red or blue, is in line with the topic of the paper and I have no comments on its content.

Reviewer 2 Report

The paper has been improved enough for is publication.

This manuscript is a resubmission of an earlier submission. The following is a list of the peer review reports and author responses from that submission.

Round 1

Reviewer 1 Report

 Review of the paper No. 520532 for Animals

The paper presented for the review contains an overview of the world literature on the nutritional usefulness of n-3 and n-6 fatty acids in poultry. The paper is written well and comprehensibly. The selection of references is correct. Before printing the paper in Animals, I suggest giving information on the recommended ratio of n6 / n3 in the human diet and n6 / n3 ratio in eggs and poultry meat, the need to modify them, the production of eggs and poultry meat with the character of functional food and their purpose, importance in the human diet, that the modified composition of the poultry diet is one of the methods of modifying the proportion of fatty acids n6 / n3. In the chapter "Improving egg quality and nutritional value of eggs", please add the paper on the effect of linseed, , soy and rapeseed oil, and above all CLA (conjugated linoleic acids) on the profile of fatty acids of egg yolk.

Other - technical notes:

Page 1: L2: omega-3 instead of Omega-3; 41-43: "essentials" or all of the n3 and n6 acids?

Page 2: L60: omega-3, omega-6, fatty acid - from lowercase; L76: after "are high" add omega-3 fatty acids.

Page 4: L168: linseed oil - LO (p<0.5); L184: 35 g / kg diet, space after 35

Page 5: L226: MDA (....) add the full name in brackets; L239: SFA instead of SAFA; L245 Sunflower oil? LNA (....) add the full name in brackets

Page 6: L258 LA (...) add the full name in brackets; L259: AA (...) add the full name in brackets.  L275-283: is rabbit also poultry ?

Page 7: L328: ALA (...) add the full name in brackets; LC-PUFA (....) add the full name in brackets; L359: GnRH, FSH, LH (no full name in brackets).

Author Response

Reviewer #1 – The paper presented for the review contains an overview of the world literature on the nutritional usefulness of n-3 and n-6 fatty acids in poultry. The paper is written well and comprehensibly. The selection of references is correct. Before printing the paper in Animals, I suggest giving information on the:  recommended ratio of n6 / n3 in the human diet and n6 / n3 ratio in eggs and poultry meat, the need to modify them, the production of eggs and poultry meat with the character of functional food and their purpose, importance in the human diet, that the modified composition of the poultry diet is one of the methods of modifying the proportion of fatty acids n6 / n3.

Response: Thanks for your valuable comment; we appreciate your support for our manuscript. Required information is inserted into the manuscript (please see lines 79-85, 266-274). Additionally, the manuscript is re-written professionally via springer nature language editing service.

In the chapter "Improving egg quality and nutritional value of eggs", please add the paper on the effect of linseed, soy and rapeseed oil, and above all CLA (conjugated linoleic acids) on the profile of fatty acids of egg yolk.

Response: Thanks for your valuable comment; additional references were inserted accordingly (Please see lines 292-306).

Other - technical notes: Page 1: L2: omega-3 instead of Omega-3; 41-43: "essentials" or all of the n3 and n6 acids?

Response: Thanks for your valuable comment; title and abstract were completely changed according to reviewer# 2 comments (Please see lines 2-3, 39-60).

Page 2: L60: omega-3, omega-6, fatty acid - from lowercase; L76: after "are high" add omega-3 fatty acids.

 Response: Thanks for your valuable comment, required changes were performed accordingly. (Please see line 61).

Page 4: L168: linseed oil - LO (p<0.5); L184: 35 g / kg diet, space after 35

Response: Thanks for your valuable comment, required changes were performed accordingly. (Please see lines 101, 214).

Page 5: L226: MDA (....) add the full name in brackets; L239: SFA instead of SAFA; L245 Sunflower oil? LNA (....) add the full name in brackets

Response: Thanks for your valuable comment, required changes were performed accordingly. (Please see lines 253,269,274).

Page 6: L258 LA (...) add the full name in brackets; L259: AA (...) add the full name in brackets. L275-283: is rabbit also poultry?

Response: Thanks for your valuable comment, required changes were performed accordingly. (Please see lines 125, 223). Sentences concerning rabbits have been limited. Since rabbit is a meat producing animal, so we now indicated only few lines as moving forward research in poulty.   

Page 7: L328: ALA (...) add the full name in brackets; LC-PUFA (....) add the full name in brackets; L359: GnRH, FSH, LH (no full name in brackets).

Response: Thanks for your valuable comment, required changes were performed accordingly. (Please see lines 90, 113, and 417-418). All abbreviations were revised throughout the manuscript.

Reviewer 2 Report

Dear Authors,

the current ms is very poorly written. I am so sorry to recommend rejection due to the poor writing.

the aim of any review is to: 1.summarise the knowledge, 2.analyse the knowledge, 3.suggest practical recommendation and 4.identify gaps for further research. 

I cannot find the components 2, 3 and 4 in the current review.

some of comments are listed below.

title: not informative. it should read: Omega 3 and omega6 fatty acids in poultry nutrition: Effect on production performance and health.

abstract: 

the abstract does not contain:
1-definition of the acids
2-importance of this review
3-overall all benefits of these acids on health and performance
4-conclusion/s (recommendation of level of inclusion for example)
5-major gaps for future research

introduction:

introduction does not reflect on the importance of reviewing o3 and o6 fatty acids.
Very poorly written.
Very short.

body of the review

line 116-125: not relevant.

paragraph3: content does not match the subheading.

the body of the text lack to the right organisation. no sufficient details were presented.

conclusion:

no gaps for the future research nor practical recommendations were presented.

Author Response

Reviewer #2 – Dear Authors, the current ms is very poorly written. I am so sorry to recommend rejection due to the poor writing. the aim of any review is to: 1.summarise the knowledge, 2.analyse the knowledge, 3.suggest practical recommendation and 4.identify gaps for further research. I cannot find the components 2, 3 and 4 in the current review. some of comments are listed below.

Response: Thanks for your comment; we appreciate your efforts and your interest in highly valuable review.The revised manuscript has been edited by professional editor of Editage (www.editage.com). We believe that we have been able to address your comments and have modified and clarified our manuscript accordingly. Our manuscript is greatly modified and supported, also it is re-written professionally by Editage editing service. Other sections were supported scientifically and rewritten (Please see lines 79-85, 129-173, 196-208, 266-274, 292-306, 308-315, 349-366, 368-371).

Title: not informative. it should read: Omega 3 and omega6 fatty acids in poultry nutrition: Effect on production performance and health.

Response: Thanks for your comment; title was changed accordingly (Please see lines 1-2).

 Abstract: the abstract does not contain: 1-definition of the acids

2-importance of this review

3-overall all benefits of these acids on health and performance

4-conclusion/s (recommendation of level of inclusion for example)

5-major gaps for future research introduction:

Response: Thanks for your comment; abstract was completely changed and re-written accordingly (Please see lines 39-60).

Introduction does not reflect on the importance of reviewing o3 and o6 fatty acids. Very poorly written. Very short. 

Response: Thanks for your comment; introduction was supported and re-written accordingly.

Body of the review line 116-125: not relevant.

Response: Thanks for your comment; Body of the review was supported and re-written accordingly.

paragraph3: content does not match the subheading. The body of the text lack to the right organisation. No sufficient details were presented.

Response: Thanks for your comment; this subheading was completely removed and its data were merged in more suitable site throughout the review.

Conclusion: No gaps for the future research or practical recommendations were presented.

Response: Thanks for your comment; conclusion was re-written in clearer and professional manner (Please see lines 233-249).

Reviewer 3 Report

The topic of the review is very interesting, and the authors have searched an important and actualized amount of bibliographic references.

The structure of the paper is well stated, and the different parts are correctly defined.

Nevertheless, the paper needs to be rewritten for publishing in a good form. it is necessary a big effort of coordination between authors to reach a good result in this review.

Some remarks about this subject:

The review is not balanced: some sections have a good and complete information, but some others like number 3 (development and metabolism) are too reduced and poor. Also, some sections are very scientific, but some others are less technical. It is necessary to balance the different sections.

Also, some ideas are repeated between sections. It is necessary to review and rewrite to avoid duplicities.

In Section 6, lines from 240 to 242 are referred to human health. It is already done (and too much in my opinion) in the introduction (lines 85 to 100).

In Section 7 (Improved meat quality) lines from 275 to 283 are referred to rabbits. It has no-sense in this paper, because the nutritional physiology of rabbits and poultry are quite different.

The acronyms need to be reviewed and used in the correct way: some of them are shown in the introduction, then in other sections are shown with the complete name. It is necessary to coordinate all sections.

The used units should be reviewed. For example, in Table 1, Authors [111] grams/kg are presented as “gm/kg” and in Authors [109] and [111] (again) are units are presented as /kg (are they g/kg?).

Author Response

Reviewer #3 – for Authors The topic of the review is very interesting, and the authors have searched an important and actualized amount of bibliographic references. The structure of the paper is well stated, and the different parts are correctly defined. Nevertheless, the paper needs to be rewritten for publishing in a good form. it is necessary a big effort of coordination between authors to reach a good result in this review.

  Response: Thanks for your valuable comment; we appreciate your support for our manuscript. Additionally, the manuscript is re-written professionally via springer nature language editing service.

Some remarks about this subject: The review is not balanced: some sections have good and complete information, but some others like number 3 (development and metabolism) are too reduced and poor. Also, some sections are very scientific, but some others are less technical. It is necessary to balance the different sections.

  Response: Thanks for your valuable comment; subheading number 3 was completely removed and its data were merged in more suitable site throughout the review according to reviewer comment. Other sections were supported scientifically and rewritten (Please see lines 79-85, 129-173, 196-208, 266-274, 292-306, 308-315, 349-366, 368-371).

Also, some ideas are repeated between sections. It is necessary to review and rewrite to avoid duplicities.

Response: Thanks for your valuable comment; the whole review is revised accordingly.

 In Section 6, lines from 240 to 242 are referred to human health. It is already done (and too much in my opinion) in the introduction (lines 85 to 100).

Response: Thanks for your valuable comment; these lines were removed, introduction was modified and rewritten accordingly.

In Section 7 (Improved meat quality) lines from 275 to 283 are referred to rabbits. It has no-sense in this paper, because the nutritional physiology of rabbits and poultry are quite different.

Response: Thanks for your valuable comment; these lines were completely removed.

The acronyms need to be reviewed and used in the correct way: some of them are shown in the introduction, and then in other sections are shown with the complete name. It is necessary to coordinate all sections.

Response: Thanks for your valuable comment; acronyms throughout review were revised and coordinate in all sections accordingly.

The used units should be reviewed. For example, in Table 1, Authors [111] grams/kg are presented as “gm/kg” and in Authors [109] and [111] (again) are units are presented as /kg (are they g/kg?).

Response: Thanks for your valuable comment; units throughout review were revised accordingly.

Thank you very much

Round 2

Reviewer 2 Report

Dear authors,

I am sorry to recommend reject of your ms.

I did not see much improvement in the text.

Regards,

Author Response

Response letter

Ref:  animals-520532

Title: " Omega 3 and omega6 fatty acids in poultry nutrition: Effect on production performance and health".

Authors: Mahmoud Alagawany, Shaaban S. Elnesr, Mayada R. Farag, Mohamed E. Abd El-Hack, Asmaa F. Khafaga, Ayman E. Taha, Ruchi Tiwari, Mohd. Iqbal Yatoo, Prakash Bhatt, Sandip Kumar Khurana and Kuldeep Dhama

Journal: Animals

Dear Prof. Dr. Betty Zhang

Editor-in-Chief

Animals

Thank you for your letter, dated July 2, 2019, and the respectable reviewers for valuable comments concerning our manuscript.

We appreciate the opportunity of submitting a revised version. Please find herewith a markedup-revised version of our article (showing the changes made in our manuscript in a yellow-green-highlighting text) and a detailed response to Editor and Reviewers’ comments. We believe that we have been able to address their comments and have modified and clarified our manuscript accordingly. We would like to thank you and the referees for your constructive criticisms and for giving us the chance to revise our manuscript.

We hope that this revised version will be acceptable for publication.

Yours sincerely,

Mahmoud Alagawany,

---------------------------

Department of Poultry,

Faculty of Agriculture, Zagazig University,

Zagazig 44511, Egypt.

E-mail address: mmalagwany@zu.edu.eg

The reviewer comments and answers

The paper has been strongly improved. It is much more balanced that the first version and only few corrections are needed:

Thanks for your valuable comment; we appreciate your support for our manuscript.

Line 105. The objective should be written in present, not in past. So “The objective of the present review is to assess the influence…”

Done

Line 291 (Improved meat quality) should be written “Improving meat quality” using the same expression that in Line 256.

Done

Lines 309 to 313 are still referred to rabbits. Please remove these comments about rabbits.

Done

In general (in many lines), the expression of ω-6 and ω-3 should be consistent in the whole paper. Sometimes you have written n-3 or n-6 while in general you write ω-6 or ω-3. Except when you write C20:5 n-3 or similar, it is recommended to write with the Greek letter ω.

Corrected

Lines 429 to 445. The conclusion should be reorganized to make it more consistent with the paper. For example, if reproduction section is the last one in the paper, please do not write their conclusions almost at the first place. Also, tell something about egg quality. Also, in line 438 please don’t write “in the present study”, as this is a review.

Thanks for your comment; conclusion was completely changed and re-written accordingly

References (ref. 96 and ref. 136 is same. Please revise) - as reflected in attached file sent by editor

Corrected

Thanks for your comment; we appreciate your efforts and your interest in highly valuable review.

Thank you very much

Reviewer 3 Report

The paper has been strongly improved. It is much more balanced that the first version. And only few corrections are needed:

Line 105. The objective should be written in present, not in past. So “The objective of the present review is to assess the influence…”

Line 291 (Improved meat quality) should be written “Improving meat quality” using the same expression that in Line 256.

Lines 309 to 313 are still referred to rabbits. Please remove these comments about rabbits.

In general (in many lines), the expression of ω-6 and ω-3 should be consistent in the whole paper. Sometimes you have written n-3 or n-6 while in general you write ω-6 or ω-3. Except when you write C20:5 n-3 or similar, it is recommended to write with the Greek letter ω.

Lines 429 to 445. The conclusion should be reorganized to make it more consistent with the paper. For example, if reproduction section is the last one in the paper, please do not write their conclusions almost at the first place. Also, tell something about egg quality. Also, in line 438 please don’t write “in the present study”, as this is a review.

Author Response

The reviewer comments and answers

The paper has been strongly improved. It is much more balanced that the first version and only few corrections are needed:

Thanks for your valuable comment; we appreciate your support for our manuscript.

Line 105. The objective should be written in present, not in past. So “The objective of the present review is to assess the influence…”

Done

Line 291 (Improved meat quality) should be written “Improving meat quality” using the same expression that in Line 256.

Done

Lines 309 to 313 are still referred to rabbits. Please remove these comments about rabbits.

Done

In general (in many lines), the expression of ω-6 and ω-3 should be consistent in the whole paper. Sometimes you have written n-3 or n-6 while in general you write ω-6 or ω-3. Except when you write C20:5 n-3 or similar, it is recommended to write with the Greek letter ω.

Corrected

Lines 429 to 445. The conclusion should be reorganized to make it more consistent with the paper. For example, if reproduction section is the last one in the paper, please do not write their conclusions almost at the first place. Also, tell something about egg quality. Also, in line 438 please don’t write “in the present study”, as this is a review.

Thanks for your comment; conclusion was completely changed and re-written accordingly

References (ref. 96 and ref. 136 is same. Please revise) - as reflected in attached file sent by editor

Corrected

Thanks for your comment; we appreciate your efforts and your interest in highly valuable review.

Thank you very much